# Influence of Frailty on Health-Related Quality of Life Trajectories in Chronic Kidney Disease Patients in India

**DOI:** 10.3390/jcm14082753

**Published:** 2025-04-17

**Authors:** Sourav Debnath, Anurag Kumar Singh, Sumit Rajotiya, Shivang Mishra, Pusparghya Pal, Preeti Raj, Hemant Bareth, Mahaveer Singh, Pratik Tripathi, Deepak Nathiya, Balvir Singh Tomar

**Affiliations:** 1Department of Pharmacy Practice, Nims University, Jaipur 303121, Rajasthan, India; sdnathme@gmail.com (S.D.); anuragkrsingh06@gmail.com (A.K.S.); sumitrajotiya199@gmail.com (S.R.); shivang.mishra23@gmail.com (S.M.); pusparghya004@gmail.com (P.P.); hemant.bareth@nimsuniversity.org (H.B.); dnathiya@nimsuniversity.org (D.N.); 2School of Health Sciences, Faculty of Biology, Medicine and Health, University of Manchester, Manchester Academic Health Science Centre, Manchester M139PL, UK; rajpreetipp@gmail.com; 3Department of Endocrinology, Nims University, Jaipur 303121, Rajasthan, India; drms.mamcmed@gmail.com; 4Department of Nephrology, Nims University, Jaipur 303121, Rajasthan, India; 5Institute of Pediatric Gastroenterology and Hepatology, Nims University, Jaipur 303121, Rajasthan, India; chancellor@nimsuniversity.org

**Keywords:** frailty, symptoms problem, dialysis, chronic kidney disease, quality of life

## Abstract

**Background**: Frailty is a critical concern for chronic kidney disease (CKD) patients, contributing to increased vulnerability to adverse health outcomes and diminished quality of life. However, there is limited research on frailty’s impact on health-related quality of life (HRQOL) among dialysis and pre-dialysis patients in the Indian context. **Methods**: This study involved participants aged 18 and above with CKD stages 3–5. Frailty was assessed using the Morley FRAIL questionnaire, and HRQOL was measured using the RAND version of the KDQOL-36 Survey. Data were analyzed with SPSS version 29, focusing on the association between frailty and HRQOL domains. **Results**: Among the 147 CKD patients, 56.46% were frail, and 43.56% were non-frail. Significant differences were noted between frail and non-frail groups in age (*p* = 0.036), CKD stages (*p* < 0.001), nutritional status (*p* < 0.001), Charlson comorbidity index (*p* < 0.001), BMI (*p* < 0.001), GFR (*p* < 0.001), CRP (*p* = 0.006), and serum albumin (*p* = 0.002). Frailty is significantly associated with lower physical (*p* < 0.001) and mental (*p* < 0.001) quality of life. Negative associations between frailty and KDQOL-36 domains, especially symptom problems, PCS, and MCS, were established. **Conclusions**: Our findings emphasize the importance of frailty screening in CKD patients. Early identification may help guide targeted strategies to support HRQOL. However, longitudinal studies are needed to assess frailty progression and the impact of potential interventions.

## 1. Background

Chronic kidney disease (CKD) is a progressive condition that diminishes kidney function over time, leading to a decreased glomerular filtration rate (GFR) [1,2]. CKD is a significant public health concern, affecting over 850 million people globally, with a prevalence of 13% worldwide [3]. The burden of CKD is higher in low- and middle-income countries, affecting 11.4% in low-income regions, 15% in middle-income areas, and 10.8% in high-income countries. CKD contributes to approximately 1.2 million deaths annually, ranking as the 12th leading cause of death globally [4,5,6].

A crucial aspect of CKD that has garnered considerable focus in recent years is frailty. Frailty is described as a geriatric syndrome, characterized by physiological decline and increased vulnerability to adverse health outcomes such as hospitalization, disease progression, surgical complications, falls, disability, and mortality [7]. Frailty is more prominent among women with chronic renal insufficiency than healthy adults and those suffering from other chronic diseases such as vasculopathy and cancers [8,9]. Studies report that frailty affects more than 60% of dialysis-dependent CKD patients, compared to about 11% of community-dwelling older adults [10,11,12].

Frailty in CKD is driven by factors such as accelerated aging, oxidative stress, inflammation, malnutrition, protein-energy wasting, and CKD-specific issues like uremia and mineral bone disease [13]. These mechanisms, compounded by a high burden of chronic comorbidities—including diabetes, hypertension, and cardiovascular diseases—can significantly impair health-related quality of life (HRQOL) in CKD patients [14,15]. Notably, HRQOL varies across CKD stages. Pre-dialysis patients generally report better HRQOL than those on dialysis or with end-stage CKD; yet, their quality of life remains substantially lower than that of healthy individuals [16].

Despite the known interplay between frailty and HRQOL, its impact on CKD patients in the Indian subcontinent remains underexplored. Existing research primarily focuses on Western populations with limited data on frailty progression, its stage-specific impact across CKD, and the absence of region-specific intervention guidelines. Additionally, gender-based differences in frailty progression and HRQOL outcomes remain poorly studied in this population. Given India’s diverse genetic predispositions, dietary habits, and socio-economic variations, understanding how frailty influences HRQOL in this setting is essential.

Hence, this study aims to examine the relationship between frailty and HRQOL trajectories in CKD patients, addressing existing knowledge gaps and providing insights that could improve patient outcomes in India.

## 2. Materials and Methods

### 2.1. Study Design and Participants

This hospital-based cross-sectional study was conducted at the Nephrology Department of the National Institute of Medical Sciences and Research, Jaipur, India, from August 2023 to March 2024. Participants aged 18–80 years with a confirmed diagnosis of stage 3–5 chronic kidney disease (CKD) based on clinical, laboratory, and histopathological evidence were eligible for enrollment. Informed consent was obtained from all participants, and the study adhered to the Declaration of Helsinki and ICMR guidelines [17,18]. The STROBE (Strengthening the Reporting of Observational Studies in Epidemiology) checklist was used to ensure transparency, accuracy, and completeness of the study [19]. Ethical approval (NIMSUR/IEC/2023/686) was granted by the Institutional Ethical Committee (IEC) of Nims University before commencing data collection.

### 2.2. Data Collection, Clinical Evaluation

Upon enrollment, participants received a comprehensive clinical evaluation that included medical history, disease duration and progression, comorbid conditions, and socio-demographic details such as age, gender, education, area of residence, occupation, smoking, alcohol, tobacco use, and dietary habits. This thorough data collection provided a holistic view of the patient’s health and potential factors influencing their kidney condition.

All biochemical parameters were analyzed in a certified clinical laboratory using standardized protocols and automated analyzers. Complete blood count (CBC), including hemoglobin (g/dL), total leukocyte count (TLC, ths/µL), and platelet count (×10^9^/L), was performed using an automated hematology analyzer (BC-6200, Mindray Bio-Medical Electronics, Shenzhen, China) based on impedance and flow cytometry principles. Renal function tests including serum creatinine (mg/dL) and urea (mg/dL) were analyzed using enzymatic colorimetric assays on the Model Elite 580 (Erba Mannheim, Mallaustraße 69, Germany), with creatinine and urea reagent kits from Erba Mannheim, Germany.

Electrolyte panels such as serum sodium (mmol/L), serum calcium (mg/dL), and serum phosphate (mg/dL) were measured using the ion-selective electrode (ISE) method for sodium and the colorimetric method for calcium and phosphate, utilizing the same analyzer and reagent kits from Erba Mannheim, Germany. Inflammatory and nutritional biomarkers including CRP (mg/L) and serum albumin (g/dL) were assessed using immunoturbidimetric and bromocresol green (BCG) colorimetric assays, respectively, on the Model Elite 580 (Erba Mannheim, Germany).

Imaging techniques such as abdominal ultrasound and KUB X-rays were used to visualize kidney architecture and detect CKD-related complications. The estimated glomerular filtration rate (GFR) was calculated using the MDRD-3 equation: GFR (mL/min/1.73 m^2^) = 175 × (Scr)^−1.54^ × (Age)^−0.203^ × (0.742 if female) × (1.212 if African American). GFR values of 30–59 mL/min/1.73 m^2^ indicate stage 3 CKD, 15–29 mL/min/1.73 m^2^ indicate stage 4, and less than 15 mL/min/1.73 m^2^ indicate stage 5 or end-stage renal disease (ESRD) [20].

### 2.3. Anthropometric and Nutritional Assessment

BMI was calculated using the formula weight (kg)/height (m^2^) and categorized according to Asia-Pacific standards. A BMI < 18.5 kg/m^2^ as underweight, 18.5 to 22.9 kg/m^2^ as normal weight, 23.0 to 24.9 kg/m^2^ as overweight, and 25.0 kg/m^2^ or higher were classified as obese [21,22]. Nutritional status was assessed using the short nutritional assessment questionnaire (SNAQ), classifying based on severity scores as follows: 0–1 points indicated well-nourished status, 2 points indicated mild malnutrition and a score of 3 or more points indicated severe malnutrition [23].

### 2.4. Comorbidities and Frailty Assessment

The Charlson comorbidity index (CCI) was employed to evaluate the comorbidities of the participants. The total scores from the CCI offer a numerical depiction of the participant’s overall comorbidity burden [24]. Frailty was assessed using the FRAIL (fatigue, resistance, aerobic capacity, illnesses, and loss of weight) questionnaire, categorizing participants as robust, pre-frail, or frail based on their frail scores of 0, 1–2, and 3, respectively [25].

### 2.5. Health-Related Quality of Life (HRQOL) Assessment Tool

HRQOL was assessed using the KDQOL-36 questionnaire [26], which includes five subscales: symptoms, effects, burden of kidney disease, physical component summary (PCS), and mental component summary (MCS). Trained staff administered the questionnaire, and responses were analyzed for internal consistency and reliability. Our study confirmed strong internal consistency, with a total Cronbach’s alpha of 0.91. Individual domain alphas were also high: symptoms (0.84), effects (0.88), burden (0.91), PCS (0.76), and MCS (0.87). These results demonstrate the questionnaire’s reliability in measuring the health status and HRQOL aspects of CKD patients.

### 2.6. Sample Size Estimation

A post hoc power analysis was conducted using G*Power 3.1.9.7 to evaluate whether the sample size was sufficient to detect a clinically meaningful difference between the two groups. A two-tailed independent t-test was used, with an effect size (Cohen’s d) of 2.06, and an alpha level of 0.05. The achieved power (1 − β) was calculated to be 1.000, indicating that the study had an extremely high probability of detecting a true effect if one existed. This suggests that the sample size was more than adequate to identify significant differences between the groups in the measured outcomes. The study employed a simple random sampling method.

### 2.7. Data Analysis

Statistical analysis was performed using SPSS version 29, while the graphical representations were generated using Microsoft Excel Software Version 1808. The Kolmogorov–Smirnov test was used to assess normality. Continuous variables were presented as means and standard deviations (SDs), or medians and interquartile ranges (IQRs), while categorical variables were presented as frequencies and percentages. Differences between frail and non-frail groups were assessed using *t*-tests, and Mann–Whitney U tests for continuous variables. For categorical variables, X2 tests were used. Hierarchical linear regression models examined the impact of frailty on PCS and MCS. Multivariate logistic regression analyzed the association between frailty and KDQOL-36 domains, with *p*-values < 0.05 considered significant. Variables identified as significant in univariate analysis were included in the multivariate logistic regression model. Collinearity was assessed using the variance inflation factor (VIF), with variables exceeding VIF > 10 excluded. Model selection was guided by the Akaike information criterion (AIC), balancing model fit and complexity. Confounding variables were selected based on prior literature, clinical relevance, and statistical considerations.

## 3. Result

### 3.1. Clinic-Demographic Characteristics of Participants

Of the 220 patients screened, 147 were included in the final analysis during the study period (Figure 1). The study cohort (N = 147) was divided into two groups: frail patients (56.46%) had a mean age of 46.37 years, while non-frail (both robust and pre-frail) patients (43.54%) had a mean age of 42.45 years. Frail individuals were older (*p* = 0.036) and predominantly male (74.4%). The majority of patients were in CKD stage 5 (76.9%), with frailty more common among dialysis patients. Hypertension was significantly associated with frailty (85.03%, *p* < 0.001), more than diabetes (13.60%), cardiovascular disease (6.80%), and glomerulonephritis (3.40%). Frail participants had higher Charlson comorbidity index scores [3(3,5) vs. 3(2,3), *p* < 0.001], reported severe illness (77.10%), and had a lower BMI (19.45 ± 3.42 kg/m^2^) (Table 1).

### 3.2. Biochemical Characteristics of Participants

Table 2 highlights significant differences observed between frail and non-frail groups across various biochemical parameters: serum urea (*p* = 0.003), serum creatinine (*p* = 0.002), GFR (*p* < 0.001), CRP (*p* = 0.006), and serum albumin (*p* = 0.002). Nutritional assessment indicated severe malnutrition in frail individuals (*p* < 0.001). Health-related quality of life (HRQOL) scores were significantly lower in frail individuals across all KDQOL domains: symptom problems (80.20 ± 7.03 vs. 51.18 ± 10.67, *p* < 0.001), effect of kidney disease (71.72 ± 11.98 vs. 38.74 ± 11.87, *p* < 0.001), burden of kidney disease (59.86 ± 18.01 vs. 14.53 ± 7.93, *p* < 0.001), PCS (43.13 ± 6.25 vs. 31.18 ± 5.45, *p* < 0.001), and MCS (49.29 ± 7.05 vs. 30.74 ± 4.60, *p* < 0.001) (Table 3, Figure 2 and Figure 3).

### 3.3. Influence of Frailty on HRQOL

Table 4 and Table 5 highlight the hierarchical predictive models assessing the impact of frailty on the physical component summary (PCS) and mental component summary (MCS) of health-related quality of life. In Table 4, Model 1 predicted PCS [F (6, 140) = 8.82, *p* < 0.001], accounting for 27.4% of the variance, with significant predictors being glomerular filtration rate (GFR) (*p* < 0.001) and Charlson comorbidity index (CCI) (*p* < 0.001). Model 2, which included frailty, explained 51.7% of the variance (R^2^ change = 24.2%, *p* < 0.001), demonstrating a strong negative association between frailty and PCS. Similarly, in Table 5, Model 1 predicted MCS [F (6, 140) = 16.56, *p* < 0.001], explaining 41.5% of the variance, with significant predictors including body mass index (BMI) (*p* = 0.003), GFR (*p* < 0.001), and CCI (*p* < 0.001). Model 2, which incorporated frailty, explained 64.8% of the variance (R^2^ change = 23.3%, *p* < 0.001), also indicating a strong negative association between frailty and MCS. Overall, frailty was found to be an independent predictor of lower physical and mental health-related quality of life.

### 3.4. Association Between Frailty and KDQOL-36 Domains

Table 6 presents the results of the multivariate logistic regression analysis, which revealed significant negative associations between frailty and several KDQOL-36 domains. Specifically, frailty was negatively associated with the symptoms/problem domain (OR: 0.843, *p* = 0.012), the physical component summary (PCS) (OR: 0.771, *p* = 0.009), and the mental component summary (MCS) (OR: 0.793, *p* = 0.017). In contrast, the effect of kidney disease (OR: 0.945, *p* = 0.370) and the burden of kidney disease (OR: 1.111, *p* = 0.074) did not show statistically significant associations with frailty. Figure 4 illustrates that as the frailty score increases, the QoL scores (SPKD, MCS, and PCS) decrease, highlighting a negative correlation between frailty and quality of life in patients. The trend lines further emphasize this decline across all QoL components.

## 4. Discussion

This study aimed to elucidate the relationship between frailty and health-related quality of life (HRQOL) among patients with chronic kidney disease (CKD) in the Indian context. Our findings revealed a significant negative correlation between frailty and various domains of HRQOL, particularly in symptom problems, physical component summary (PCS), and mental component summary (MCS) as assessed by the KDQOL-36.

Our study found a frailty prevalence of 56.46% among CKD patients, consistent with prior research indicating higher frailty rates in this population compared to the general elderly population. For instance, studies by Johansen et al. [11] and Collard et al. [12] reported frailty prevalence rates of over 60% in dialysis-dependent CKD patients. The advanced age of our frail participants aligns with other studies suggesting that older CKD patients are more prone to frailty due to physiological decline and multiple comorbidities [8].

In our cohort, hypertension was more prevalent among frail patients, supporting findings by Lee et al. [14] that CKD patients with hypertension and diabetes are more susceptible to frailty. However, we did not analyze the direct association between hypertension and frailty in this study. Moreover, the Charlson comorbidity index (CCI) scores were notably higher among frail individuals, highlighting the cumulative burden of comorbid conditions in exacerbating frailty. Future longitudinal studies are needed to explore potential causal relationships between hypertension and frailty.

Contrary to some studies suggesting higher frailty rates among women, our study observed a higher prevalence of frailty among men [11]. However, this difference was not statistically significant (*p* = 0.557). This discrepancy might be attributed to our study’s small sample size or cultural factors affecting healthcare access and reporting among women in India. Further research with larger, more diverse cohorts is needed to explore these gender differences.

In the context of the biochemical milieu, frail patients exhibited lower serum albumin and GFR levels, alongside higher C-reactive protein (CRP) levels, indicating a state of chronic inflammation and malnutrition. Hypoalbuminemia, a widely recognized biomarker of poor nutritional status and a significant predictor of adverse outcomes in CKD patients, often results from protein-energy wasting, leading to cachexia. Studies suggest that hypoalbuminemia contributes to frailty progression through mechanisms such as maintaining oncotic pressure, serving as a carrier protein, and exerting antioxidant and anti-inflammatory effects [27]. Impaired albumin functions cause fluid imbalances, increased oxidative stress, and heightened inflammation, exacerbating physical decline and vulnerability [28]. This condition is mainly seen in patients on maintenance hemodialysis due to severe albumin loss during dialysis [29]. Additionally, studies have shown that low albumin levels in CKD increase the risk of sarcopenia, a hallmark of frailty [30,31]. Our study reported severe malnutrition in frail patients, assessed by the SNAQ questionnaire. However, as our study was cross-sectional, it cannot establish a causal relationship between hypoalbuminemia and frailty progression. These findings underscore the importance of regularly assessing nutritional status and conducting routine serum albumin tests, especially in advanced CKD patients. Future longitudinal studies are needed to determine the directionality of this association.

Higher CRP levels were linked with greater mortality risk in frail older adults [27]. Wassel et al. indicated that higher serum CRP levels were associated with reduced survival time, particularly in males [32]. Consistent with the study by Nakazato et al. [33], our study did not observe a statistically significant difference in hemoglobin levels between frail and non-frail patients (*p* = 0.069). However, Ahmed et al. [34] reported a significant association between frailty and low hemoglobin levels. These findings suggest that the relationship between anemia and frailty may differ across populations, highlighting the need for further investigation.

Furthermore, frail patients with anemia exhibited lower GFR compared to pre-frail patients, suggesting an association between reduced kidney function and frailty in CKD. However, the strength of this association appeared to be influenced by the method used to estimate GFR. Prior studies indicate that the relationship between frailty and kidney function is attenuated when GFR is calculated solely using serum creatinine (SCr) rather than a combination of SCr and cystatin C (CysC) [35]. In the present study, GFR was estimated using serum creatinine alone, which may have influenced the observed findings. Moreover, while our study identified an association between reduced GFR and frailty, we did not analyze GFR as an independent predictor in the logistic regression model. Future research should incorporate regression analyses to evaluate the specific contribution of GFR to frailty, which could provide a clearer understanding of its role in frailty progression among CKD patients.

Concerning frailty and HRQOL, MLR analysis showed a strong negative association between frailty and all KDQOL-36 domains, except the burden of kidney disease and the effect of kidney disease domains. The ’symptom problem’ domain from the KDQOL-36 questionnaire threw up as a peculiar concern followed by the PCS and MCS domains. This is consistent with the current literature and frailty is independently associated with a high symptom burden and poor HRQOL among patients with CKD [36].

In the hierarchical linear regression model, frailty independently explained variance in lower PCS and MCS scores. In the final models, frailty was a more important predictor of HRQOL than CCI and GFR, showing its independent impact on both the physical and mental health components of CKD patients. Nevertheless, low GFR and elevated CCI scores emerged as independent predictors, indicating that disease severity is linked to diminished quality of life in CKD patients, as established in previous studies [14,35]. There have been studies that confirmed the association between frailty and comorbidities related to physical and psychological aspects among different geographic populations [15]. Therefore, interventions targeting these modifiable risk factors may be more effective in reducing the frailty burden in CKD patients rather than focusing on kidney function alone. Although frailty is prevalent in CKD patients, it is not directly assessed by the KDQOL-36, which limits the integrated analysis between physical functioning and quality of life. This is likely because the KDQOL-36 primarily assesses health-related quality of life (HRQoL) in CKD patients by focusing on physical and mental health, symptom burden, and the impact of kidney disease on daily activities. Future research should consider integrating frailty-specific assessments alongside the KDQOL-36 to provide a more comprehensive evaluation of HRQoL in CKD patients. To the best of our knowledge, no prior study from India has examined the association between frailty and HRQOL in CKD patients with or without dialysis.

Our study has multiple limitations. First, the sample size was relatively small and derived from a cross-sectional design, which did not permit the establishment of a causal inference relationship between frailty and poor HRQOL. Second, self-report measures are inherently predisposed to bias. Third, conducting the study in a single center may limit the generalizability of the results. Fourth, multiple analyses were conducted without formal corrections, which may increase the Type I error risk. However, as our analysis was primarily hypothesis-driven rather than exploratory, the impact of false positives is mitigated. Despite these limitations, this cross-sectional study among the Indian population provides a comprehensive examination of the relationship between frailty and HRQOL in CKD patients, considering a wide range of clinical and biochemical parameters. The use of well-established tools like KDQOL-36 would ensure a more reliable measurement of the HRQOL for both pre-dialysis and dialysis patients. The association between frailty and poorer HRQOL highlights the importance of considering frailty in clinical practice, although longitudinal studies are necessary to evaluate the effects of interventions. Given the significant influence of nutrition on kidney disease progression and frailty development, further research is needed to explore its impact and the potential benefits of nutritional interventions.

## 5. Conclusions

Our study showed that frail patients had significantly lower scores in the symptom burden, PCS, and MCS domains of HRQOL. Frailty was identified as a significant independent factor associated with low physical and mental health in CKD patients. As of now, there is no established treatment for frailty. Healthcare providers should prioritize comprehensive assessments of frailty status alongside routine CKD management protocols. Future studies should assess whether interventions targeting modifiable risk factors, such as nutrition and physical activity, can mitigate the adverse effects of frailty on quality of life.

## Figures and Tables

**Figure 1 jcm-14-02753-f001:**
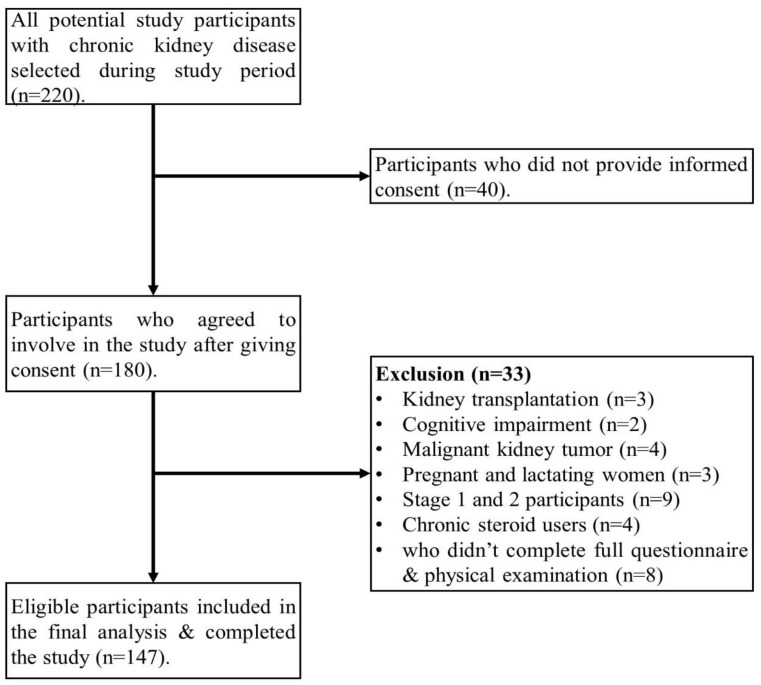
Flow chart representing participants’ selection.

**Figure 2 jcm-14-02753-f002:**
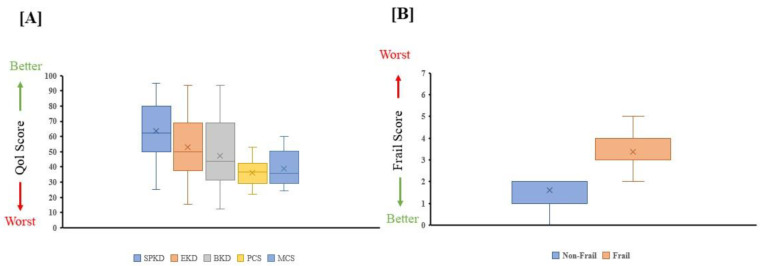
(**A**) Kidney Disease Quality of Life (KDQOL-36) questionnaire scores for five health-related quality of life domains: symptom problem kidney disease (SPKD), effect kidney disease (EKD), burden kidney disease (BKD), physical component summary (PCS), and mental component summary (MCS). (**B**) Frailty index scores for the non-frail and frail groups. Both (**A**,**B**) are represented as box plots showing quartiles, 5th and 95th percentiles (whiskers), and minimum and maximum scores.

**Figure 3 jcm-14-02753-f003:**
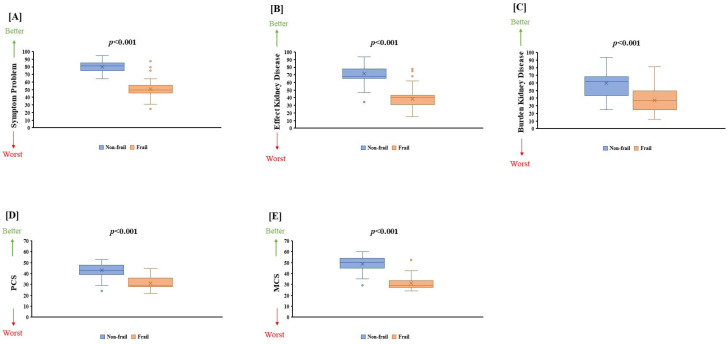
The comparison of (**A**) symptom problem kidney disease (SPKD), (**B**) effect kidney disease (EKD), (**C**) burden kidney disease (BKD), (**D**) physical component summary (PCS), and (**E**) mental component summary (MCS) between non-frail and frail groups are shown.

**Figure 4 jcm-14-02753-f004:**
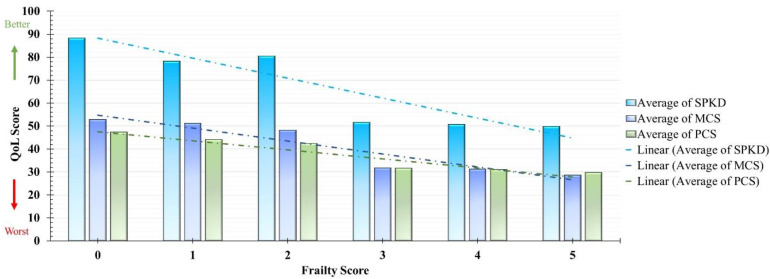
The association between KDQOL-36 domains and frailty score.

**Table 1 jcm-14-02753-t001:** Clinic-demographic characteristics of participants enrolled in the study (N = 147).

Variables	Frail (n = 83, 56.46%)	Non-Frail (n = 64, 43.54%)	Total (N = 147)	*p*-Value *
**Age(year),** mean ± SD	46.37 ± 13.89	42.45 ± 12.06	44.67 ± 13.23	**0.036**
**Gender,** n (%)				0.557
Male	62 (74.7)	45 (70.31)	107 (72.8)	
Female	21 (25.3)	19 (29.69)	40 (27.2)	
**Education**, n (%)				0.816
Illiterate	15 (18.07)	13 (20.31)	28 (19.0)	
Primary school	28 (33.73)	17 (26.56)	45 (30.6)	
High school	25 (30.12)	21 (32.81)	46 (31.3)	
Graduate	13 (15.66)	12 (18.75)	25 (17.0)	
Postgraduate	2 (2.40)	1 (1.56)	3 (2.0)	
**Area of living**, n (%)				0.995
Rural	70 (84.33)	54 (84.37)	124 (84.4)	
Urban	13 (15.66)	10 (15.62)	23 (15.6)	
**CKD stages**, n (%)				**<0.001**
**Pre-dialysis**				
Stage-3	1 (1.20)	15 (23.43)	16 (10.9)	
Stage-4	5 (6.02)	13 (20.31)	18 (12.2)	
**Dialysis**				
Stage-5	77 (92.77)	36 (56.25)	113 (76.9)	
**Diet,** n (%)				0.668
Veg	27 (32.53)	23 (35.93)	50 (34.0)	
Both	56 (67.46)	41 (64.06)	97 (66.0)	
**Occupation**				0.197
Employed	29 (34.93)	16 (25.0)	45 (30.6)	
Unemployed	54 (65.06)	48 (75.0)	102 (69.4)	
**Smoking**, n (%)				0.398
Current	19 (22.89)	12 (18.75)	31 (21.1)	
Former	27 (32.53)	19 (29.68)	46 (31.3)	
Never	37 (44.57)	33 (51.56)	70 (47.6)	
**Tobacco**, n (%)				0.777
Current	7 (8.43)	7 (10.93)	14 (9.5)	
Former	12 (14.45)	8 (12.5)	20 (13.6)	
Never	64 (77.10)	49 (76.56)	113 (76.9)	
**Alcohol**, n (%)				0.901
Current	15 (18.07)	9 (14.06)	24 (16.3)	
Former	18 (21.68)	18 (28.12)	36 (24.5)	
Never	50 (60.24)	37 (57.81)	87 (59.2)	
**CCI, Median** [IQR (Q1–Q3)]	3(3,5)	3 (2,3)	4 (3,5)	**<0.001**
**CCI Category**, n (%)				**<0.001**
Mildly ill	0 (0)	17 (26.56)	17 (11.6)	
Moderately ill	19 (22.89)	44 (68.75)	63 (42.9)	
Severely ill	64 (77.10)	3 (4.68)	67 (45.6)	
**Co-morbidity**, n (%)				**<0.001**
Hypertension	80 (96.38)	44 (68.75)	124 (84.35)	
Diabetes	11 (13.25)	9 (14.06)	20 (13.60)	
Cardiovascular diseases	3 (3.61)	5 (7.81)	10 (6.80)	
Glomerulonephritis	3 (3.61)	2 (3.12)	5 (3.40)	
**BMI** (kg/m^2^), mean ± SD	19.45 ± 3.42	21.99 ± 4.05	20.56 ± 3.90	**<0.001**
Underweight, n (%)	36 (73.47)	13 (26.53)	49 (33.3)	
Normal Weight, n (%)	41 (63.08)	24 (36.92)	65 (44.2)	
Overweight, n (%)	0	15 (100.0)	15 (10.2)	
Obese, n (%)	6 (33.35)	12 (66.65)	18 (12.2)	

All the values are presented as frequency (%) or mean and standard deviation (SD) otherwise stated. IQR—interquartile range; CCI—Charlson comorbidity index; BMI—body mass index; CKD—chronic kidney disease. * *p*-value considered statistically significant at < 0.01. The *p*-values of significant variables are in bold.

**Table 2 jcm-14-02753-t002:** Biochemical parameters of patients enrolled in the study (N = 147).

Lab Parameters	Frail (n = 83)	Non-Frail (n = 64)	Total (n = 147)	*p*-Value *
HB (g/dL)	8.27 ± 1.55	8.74 ± 1.48	8.48 ± 1.53	0.069
TLC (ths/μL)	9.58 ± 6.84	8.55 ± 3.77	9.13 ± 5.72	0.281
Serum Urea (mg/dL)	150.67 ± 63.69	119.17 ± 60.16	136.95± 63.69	**0.003**
Serum Creatinine (mg/dL)	8.30 ± 3.19	6.48 ± 3.83	7.51 ± 3.59	**0.002**
GFR (mL/min/1.73m^2^)	9.13 ± 5.76	18.72 ± 15.61	13.31 ± 12.27	**<0.001**
Serum Calcium (mg/dL)	8.14 ± 1.26	8.31 ± 1.27	8.21 ± 1.27	0.413
Serum Sodium (mmol/L)	137.38 ± 4.17	137.80 ± 3.34	137.56 ± 3.82	0.508
Serum Phosphate (mg/dL)	5.80 ± 2.25	5.62 ± 2.05	5.72 ± 2.16	0.635
CRP (mg/dL)	65.79 ± 69.00	37.80 ± 46.48	53.61 ± 61.65	**0.006**
Serum Albumin	2.76 ± 0.74	3.16 ± 0.73	2.93 ± 0.76	**0.002**

All the values are presented as frequency (%) or mean and standard deviation (SD) otherwise stated. CRP—C-reactive protein; GFR—glomerular filtration rate; * *p*-value considered statistically significant at <0.05 or <0.01. The *p*-values of significant variables are in bold.

**Table 3 jcm-14-02753-t003:** Comparison of nutritional status and quality of life domains between frail and non-frail groups.

	Frail (n = 83)	Non-Frail (n = 64)	Total (n = 147)	*p*-Value *
**SNAQ**				**<0.001**
No Intervention	0 (0)	8 (100.0)	8 (5.4)	
Moderate Malnourished	7 (8.43)	34 (53.12)	41 (27.60)	
Severely Malnourished	76 (91.56)	22 (34.37)	98 (66.7)	
**KDQOL-36**				
Symptom Problems	51.18 ± 10.67	80.20 ± 7.03	63.82 ± 17.13	**<0.001**
Effect Kidney Disease	38.74 ± 11.87	71.72 ± 11.98	53.10 ± 20.26	**<0.001**
Burden Kidney Disease	14.53 ± 7.93	59.86 ± 18.01	34.26 ± 26.15	**<0.001**
PCS	31.18 ± 5.45	43.13 ± 6.25	36.38 ± 8.30	**<0.001**
MCS	30.74 ± 4.60	49.29 ± 7.05	38.81 ± 10.88	**<0.001**

All the values are presented as frequency (%) or mean and standard deviation (SD) otherwise stated. KDQOL—kidney disease and quality of life; PCS—physical component summary; MCS—mental component summary; SNAQ—short nutritional assessment questionnaire. * *p*-value considered statistically significant at <0.05 or <0.01. The *p*-values of significant variables are in bold.

**Table 4 jcm-14-02753-t004:** Hierarchical linear regression analysis for the influence of frailty on PCS.

	Predictors	ꞵ	t	95% CI	*p*-Value
Model 1 (R^2^ = 0.274)	Age	0.063	0.786	−0.060 to 0.139	0.433
	Gender	−0.017	−0.228	−2.956 to 2.344	0.820
	BMI	0.112	1.491	−0.078 to 0.553	0.138
	GFR	0.278	3.678	0.087 to 0.288	**<0.001**
	CRP	−0.142	1.921	−0.039 to 0.001	0.057
	Comorbidity (CCI)	−0.0294	−3.577	−1.653 to −0.476	**<0.001**
Model 2 (R^2^= 0.517, R^2^ change = 0.242)	Age	0.122	1.812	−0.007 to 0.161	0.073
	Gender	−0.033	−0.543	−2.822 to 1.606	0.386
	BMI	−0.102	−1.495	−0.500 to 0.069	0.137
	GFR	0.023	0.320	0.081 to 0.112	**0.027**
	CRP	−0.087	−1.338	−0.029 to 0.006	0.056
	Comorbidity (CCI)	−0.002	−0.031	−0.560 to 0.542	0.620
	Frailty	−0.474	−5.618	−4.845 to −2.322	**<0.001**

PCS—physical component summary; CI—confidence interval; CCI—Charlson comorbidity index; BMI—body mass index; CRP—C-reactive protein; GFR—glomerular filtration rate. *p*-value considered statistically significant at <0.05 or <0.01. The *p*-values of significant variables are in bold.

**Table 5 jcm-14-02753-t005:** Hierarchical linear regression analysis for the influence of frailty on MCS.

	Predictors	ꞵ	t	95% CI	*p*-Value
Model 1 (R^2^ = 0.415)	Age	0.07	0.101	−0.111 to 0.123	0.920
	Gender	0.056	0.865	−1.754 to 4.481	0.389
	BMI	0.208	3.078	0.207 to 0.949	0.098
	GFR	0.326	4.811	0.170 to 0.406	**<0.001**
	CRP	−0.036	−0.538	−0.029 to 0.017	0.592
	Comorbidity (CCI)	−0.384	−5.201	−2.513 to −1.129	**<0.001**
Model 2 (R^2^= 0.648, R^2^ change= 0.233)	Age	0.058	1.011	−0.046 to 0.142	0.249
	Gender	0.034	0.667	−1.639 to 3.308	0.735
	BMI	−0.001	−0.021	−0.321 to 0.314	0.200
	GFR	0.079	1.282	−0.038 to 0.177	**0.001**
	CRP	0.002	0.031	−0.019 to 0.019	0.754
	Comorbidity (CCI)	−0.098	−1.488	−1.078 to 0.152	0.058
	Frailty	−0.484	−6.733	−6.206 to 3.308	**<0.001**

MCS—mental component summary; CI—confidence interval; BMI—body mass index; CRP—C-reactive protein; GFR—glomerular filtration rate. *p*-value considered statistically significant at <0.05 or <0.01. The *p*-values of significant variables are in bold.

**Table 6 jcm-14-02753-t006:** Association of frailty with KDQOL-36 domains.

Variables	ꞵ	OR	95% CI	*p*-Value
Symptom Problems	−0.170	0.843	0.739–0.963	**0.012**
Effect Kidney Disease	−0.057	0.945	0.834–1.070	0.370
Burden Kidney Disease	0.106	1.111	0.990–1.248	0.074
PCS	−0.261	0.771	0.633–0.938	**0.009**
MCS	−0.232	0.793	0.655–0.960	**0.017**

The reference category is non-frail. OR—odds ratio; CI—confidence interval. *p*-value considered statistically significant at <0.05 or <0.01. The *p*-values of significant variables are in bold.

## Data Availability

The data supporting the findings of this study are available from the corresponding author upon reasonable request.

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
