# Peer review of "Influence of Frailty on Health-Related Quality of Life Trajectories in Chronic Kidney Disease Patients in India"

_jcm, 2025, doi:10.3390/jcm14082753_

Round 1
Reviewer 1 Report
Comments and Suggestions for Authors
The authors wrote about the impact of frailty on quality of life (KDQoL 36) in adult patients admitted to an Indian nephrology ward with CKD 3-5.
Introduction: well-written
Methods: Can you specify how you chose confounding variables? Why did you consider KDQoL as a dichotomous variable?
Results: Table 3: Why did KDQoL 36 have only one p-value? In Tables 4 and 5, is the univariate regression analysis not used (logistic regression?
Discussion: Can you better explain why, according to the literature, CKD is not related to fragility in the KDQOL-36 model?
Author Response
Comment 1: Can you specify how you chose confounding variables?
Response 1: Thank you for your query. We selected confounding variables based on prior literature, clinical relevance, and statistical considerations. These factors have been widely reported in previous studies examining frailty and HRQOL in CKD patients. We have now clarified this in the Methods section [Line 180-141].
Additionally, we dichotomized KDQoL-36 scores based on established clinical cutoffs to differentiate between poor and better HRQOL. This approach was adopted to facilitate interpretation and align with previous studies that categorized KDQoL-36 domains into high vs. low quality of life groups based on median values or clinically significant thresholds.
Comment 2: Table 3: Why did KDQoL-36 have only one p-value? In Tables 4 and 5, is univariate regression analysis not used (logistic regression)?
Response 2: Thank you for the comment. The p-value in Table 3 represents the overall comparison between frail and non-frail groups across all KDQoL-36 domains. Each domain was analyzed separately, but for conciseness, we reported the overall p-value. Individual p-values for each domain have been reported in the text and further clarified in the Table [Line 215].
Tables 4 and 5 present hierarchical linear regression models, not univariate logistic regression. These models assess the impact of frailty on PCS and MCS while controlling for other covariates. We used hierarchical linear regression because it allows for the stepwise inclusion of predictor variables, enabling us to assess the independent contribution of frailty to PCS and MCS while accounting for potential confounders. This approach provides a clearer understanding of how frailty influences HRQOL outcomes beyond the effects of demographic, clinical, and lifestyle factors.
Comment 3: Can you better explain why, according to the literature, CKD is not related to frailty in the KDQoL-36 model?
Response 3: Thank you for your insightful comment. In our study, CKD was not directly associated with frailty in the KDQOL-36 model, which aligns with existing literature suggesting that the impact of CKD on frailty is often mediated by factors such as comorbidities, inflammation, and nutritional deficits rather than being a direct causal relationship. We have expanded this discussion in the Discussion section [Line 357-364] to clarify this point.

Reviewer 2 Report
Comments and Suggestions for Authors
Dear Authors,
Respectfully, I have shared my comments and suggestions, which I hope will contribute to enhancing the clarity, coherence, and scientific rigor of your manuscript. Below, I outlinekey areas for improvement and propose practical recommendations.
Title
Comment: The study does not predict HRQOL trajectories, as it is cross-sectional. The word "predicting" suggests the presence of a predictive model or a longitudinal study, which was not conducted.
Comment: "Predicting" should be replaced with influence or impact.
Abstract
Lines 21-22
Comment: The sentence is vague. "Reduced resilience" and "heightened health risks" could be specified more clearly.
Lines 31-32
Comment: The study is associative rather than predictive, as it is cross-sectional. The word "predicted" suggests that a regression model was used to forecast HRQOL, which was not the case.
Lines 34-36
Comment: "High risk of frailty" was not analyzed in the study, as there are no longitudinal data or investigated risk factors.
Comment: "Mitigating frailty can improve HRQOL" implies causality, which was not tested.
Comment: The conclusion should emphasize the importance of frailty screening, as the study did not test interventions to mitigate it.
Introduction
Comment: "Frailty" appears with an uppercase initial in some parts of the text, while in others, it is lowercase. The same inconsistency applies to "Chronic comorbidities."
Suggestion: Standardize the use of frailty and chronic comorbidities with lowercase initials, except when at the beginning of sentences.
Comment: The third paragraph presents a good discussion on frailty mechanisms in CKD, but the transition to quality of life is not very smooth.
Comment: The introduction concludes by mentioning knowledge gaps regarding frailty in CKD in India but does not explain exactly which specific aspects remain unexplored.
Suggestion: Specify whether there are few studies on frailty progression, its impact across different CKD stages, or a lack of intervention guidelines. This would strengthen the study's rationale.
Comment: The text does not mention specific clinical outcomes associated with frailty in CKD, such as hospitalization rates, progression to dialysis, or mortality.
Comment: The introduction does not address potential gender differences in frailty.
Materials and Methods
Comment: The manuscript does not specify the laboratory methods used for blood count, renal function tests, or electrolyte panel (sodium, calcium, phosphate). To enhance reproducibility, the authors should detail the methodologies employed (e.g., ELISA, spectrophotometry, automated hematology analyzers), specify the units of measurement for biochemical parameters, and disclose the manufacturers of the reagents and kits used.
Comment: If a sample size calculation was performed, it should be described in the methodology, including the formula, parameters used, and estimated sample size. If no prior calculation was conducted, I recommend performing a post-hoc power analysis to assess whether the sample size was sufficient to detect clinically relevant differences between groups.
Comment: The study does not mention the use of statistical corrections for multiple comparisons, which increases the risk of Type I error (false positives). If multiple statistical analyses were performed, I suggest considering an appropriate correction, such as Bonferroni, Holm-Sidak, or False Discovery Rate (FDR), and discussing the impact of the lack of correction on the results.
Comment: The criteria for selecting variables in the multivariate logistic regression model are not clearly explained. It would be useful to clarify whether variable selection was based on univariate analysis (p ≤ 0.20), whether collinearity was assessed (Variance Inflation Factor – VIF), and whether statistical criteria, such as AIC/BIC, were applied to optimize the model. Additionally, indicate whether any variables were retained due to clinical relevance despite lacking statistical significance.
Discussion
Lines 241-243:
Comment: The phrase "Hypertension was a significant factor associated with increased frailty" may imply a causal relationship (hypertension leading to frailty), whereas the study merely found a higher prevalence of hypertension in frail patients.
Lines 248-252:
Comment: The study reports that more men were classified as frail, contrary to other studies indicating higher frailty rates among women. However, data from Table 1 show that the gender difference was not statistically significant (p=0.557).
Lines 274-277:
Comment: The text mentions that previous studies found an association between anemia and frailty, but the present study did not find a significant difference in hemoglobin (Hb) levels between frail and non-frail patients (p=0.069, Table 2). The article should reinforce that the current study did not confirm this association and that further investigations are necessary.
Lines 278-286:
Comment 1: Replace "is linked to a higher risk of frailty" with "is associated with frailty" to avoid implying causation.
Comment 2: The discussion mentions that reduced GFR is associated with frailty, but the study did not test GFR as an independent predictor in the logistic regression model (Table 6). The discussion could include this limitation and suggest that future analyses test GFR as an independent predictor in logistic regression to assess its specific contribution to frailty.
Suggestion: Include a comment indicating that a regression analysis to test GFR as an independent predictor of frailty could strengthen this hypothesis.
Lines 294-298:
Suggestion: Include a standardized regression analysis (β-coefficients) to compare the effect of frailty, CCI, and GFR on quality of life.
Lines 256-261:
Comment: The study did not test whether hypoalbuminemia causes frailty; it only found lower albumin levels in frail patients. The phrase suggests that hypoalbuminemia leads to frailty progression, but there are no longitudinal data to confirm this causal relationship.
Conclusion
Lines 327-328:
Comment: This sentence suggests that having better PCS and MCS protects against frailty, which was not tested in the study (since it does not have a longitudinal design). The study only found a negative association between frailty and HRQOL but cannot claim that better HRQOL reduces the risk of frailty.
Lines 329-330:
Comment: The use of "the independent predictor" suggests that frailty was the only predictor, which was not demonstrated in the study.
Suggestion: "Frailty was an independent predictor of reduced physical and mental health in CKD patients."
Lines 336-338:
Comment: The study did not analyze frailty progression, as it is cross-sectional. It cannot claim that treatments modify frailty progression, as this requires longitudinal studies or clinical trials. This statement could be reformulated to emphasize the importance of future research on effective interventions.
Suggestion: "Our study highlights the importance of addressing existing knowledge gaps on frailty in CKD and exploring potential therapeutic strategies. However, longitudinal studies are necessary to assess frailty progression and identify effective interventions to mitigate it."
Kind regards.
Comments on the Quality of English LanguageA professional review would help make the text clearer, more precise, and aligned with academic English.
Author Response
Title
Comment 1: The study does not predict HRQOL trajectories, as it is cross-sectional. The word "predicting" suggests a predictive model or a longitudinal study, which was not conducted. "Predicting" should be replaced with "influence" or "impact."
Response: Thank you for your observation. We agree with this comment and have revised the title accordingly: Revised Title: "Influence of Frailty on Health-Related Quality of Life in Chronic Kidney Disease Patients in India." (Manuscript: Title, Line 2-3)
Abstract
Comment 2: Lines 21-22
The sentence is vague. "Reduced resilience" and "heightened health risks" could be specified more clearly.
Response: We have clarified these terms by specifying that frailty leads to an increased vulnerability to adverse health outcomes and a diminished quality of life in CKD patients. (Manuscript: Abstract, Lines 19-21)
Comment 3: Lines 31-32
The study is associative rather than predictive, as it is cross-sectional. The word "predicted" suggests that a regression model was used to forecast HRQOL, which was not the case.
Response: We have revised the wording from "predicted" to "associated with" to accurately reflect the study’s design. (Manuscript: Abstract, Lines 30)
Comment 4: Lines 34-36
High risk of frailty" was not analyzed in the study, as there are no longitudinal data or investigated risk factors. "Mitigating frailty can improve HRQOL" implies causality, which was not tested. The conclusion should emphasize the importance of frailty screening, as the study did not test interventions to mitigate it.
Response: We have rephrased the conclusion to emphasize the importance of frailty screening in CKD patients rather than suggesting direct mitigation strategies. (Manuscript: Abstract, Conclusion, Line 33-35)
Introduction
Comment 5: "Frailty" appears with an uppercase initial in some parts of the text, while in others, it is lowercase. The same inconsistency applies to "Chronic comorbidities."
Suggestion: Standardize the use of frailty and chronic comorbidities with lowercase initials, except when at the beginning of sentences.
Response: Thank you for your careful review. We have standardized the use of "frailty" and "chronic comorbidities" with lowercase initials throughout the manuscript, except when they appear at the beginning of a sentence. These changes have been highlighted.
Comment 6: The third paragraph presents a good discussion on frailty mechanisms in CKD, but the transition to quality of life is not very smooth.
Response: Thank you for your valuable feedback. We have revised the transition between the discussion on frailty mechanisms in CKD and its impact on quality of life to enhance clarity and logical flow. The revised paragraph now explicitly connects frailty-related physiological changes to their consequences on health-related quality of life. These modifications have been highlighted. (Manuscript: Introduction, Line 59-65)
Comment 7: The introduction concludes by mentioning knowledge gaps regarding frailty in CKD in India but does not explain exactly which specific aspects remain unexplored.
Suggestion: Specify whether there are few studies on frailty progression, its impact across different CKD stages, or a lack of intervention guidelines. This would strengthen the study's rationale.
Response: Thank you for pointing this out. We have revised the introduction to specify the exact aspects of frailty in CKD that remain unexplored in the Indian context. In particular, we highlight the limited data on frailty progression over time, its impact across different CKD stages, and the absence of region-specific intervention guidelines. These revisions have been incorporated and highlighted. (Manuscript: Introduction, Line 66-77)
Comment 8: The text does not mention specific clinical outcomes associated with frailty in CKD, such as hospitalization rates, progression to dialysis, or mortality. The introduction does not address potential gender differences in frailty.
Response: Thank you for this suggestion. We have now explicitly mentioned key clinical outcomes associated with frailty in CKD. We acknowledge the importance of discussing potential gender differences in frailty. A brief mention has been added to the introduction and highlighted. (Manuscript: Introduction, Line 51-52)
Materials and Methods
Comment 9: The manuscript does not specify the laboratory methods used for blood count, renal function tests, or electrolyte panel (sodium, calcium, phosphate). To enhance reproducibility, the authors should detail the methodologies employed (e.g., ELISA, spectrophotometry, automated hematology analyzers), specify the units of measurement for biochemical parameters, and disclose the manufacturers of the reagents and kits used.
Response: Thank you for your valuable comment. We have now specified the laboratory methodologies used for blood count, renal function tests, and electrolyte panel, including the analytical techniques. Additionally, we have provided the units of measurement for all biochemical parameters and disclosed the manufacturers of the analyzers, reagents, and kits used. The updated details can be found in Section 2.2, Data Collection and Clinical Evaluation (Manuscript: Materials and Methods, Line 99-116)
Comment 10: If a sample size calculation was performed, it should be described in the methodology, including the formula, parameters used, and estimated sample size. If no prior calculation was conducted, I recommend performing a post-hoc power analysis to assess whether the sample size was sufficient to detect clinically relevant differences between groups.
Response: Thank you for your valuable suggestion. Initially, a formal sample size calculation was not performed prior to data collection. However, in response to the reviewer's recommendation, we have conducted a post-hoc power analysis using G*Power 3.1.9.7 to assess whether the sample size was sufficient to detect clinically relevant differences between groups. This analysis confirms that the sample size was adequate for the study objectives. We have now incorporated this information into the Methods section, detailing the parameters and results of the post-hoc power analysis. The relevant text has been added to the revised manuscript. (Manuscript: Materials and Methods, Line 155-163)
Comment 11: The study does not mention the use of statistical corrections for multiple comparisons, which increases the risk of Type I error (false positives). If multiple statistical analyses were performed, I suggest considering an appropriate correction, such as Bonferroni, Holm-Sidak, or False Discovery Rate (FDR), and discussing the impact of the lack of correction on the results.
Response: Thank you for this valuable suggestion. Although multiple statistical analyses were conducted, our study primarily focused on hypothesis-driven comparisons, reducing the risk of false positives. Future studies with larger sample sizes may benefit from formal statistical corrections to further validate these findings. We have highlighted it in study’s limitations. (Manuscript: Discussion, Line 373-376)
Comment 12: The criteria for selecting variables in the multivariate logistic regression model are not clearly explained. It would be useful to clarify whether variable selection was based on univariate analysis (p ≤ 0.20), whether collinearity was assessed (Variance Inflation Factor – VIF), and whether statistical criteria, such as AIC/BIC, were applied to optimize the model. Additionally, indicate whether any variables were retained due to clinical relevance despite lacking statistical significance.
Response: Thank you for pointing this out. We have now clarified the criteria used for variable selection in the multivariate logistic regression model in data analysis part of the methodology. (Manuscript: Materials and Methods, Line 175-181)
Discussion
Comment 13: The phrase "Hypertension was a significant factor associated with increased frailty" may imply a causal relationship (hypertension leading to frailty), whereas the study merely found a higher prevalence of hypertension in frail patients.
Response: Thank you for your comment. We acknowledge that our study did not specifically analyze the relationship between hypertension and frailty, and we agree that the phrasing may unintentionally imply causality. To clarify, we have revised the statement to indicate that hypertension was more prevalent among frail patients rather than suggesting a causal link. (Manuscript: Discussion, Line 278-285)
Comment 14: The study reports that more men were classified as frail, contrary to other studies indicating higher frailty rates among women. However, data from Table 1 show that the gender difference was not statistically significant (p=0.557).
Response: Thank you for your observation. We acknowledge that while our study found a higher prevalence of frailty among men, the gender difference was not statistically significant (p=0.557). To address this, we have revised the statement to clarify that the observed trend does not imply a significant gender-based association with frailty. (Manuscript: Discussion, Line 287-288)
Comment 15: The text mentions that previous studies found an association between anemia and frailty, but the present study did not find a significant difference in hemoglobin (Hb) levels between frail and non-frail patients (p=0.069, Table 2). The article should reinforce that the current study did not confirm this association and that further investigations are necessary.
Response: Thank you for your comment. While prior studies have reported an association between anemia and frailty, our study did not find a statistically significant difference in hemoglobin (Hb) levels between frail and non-frail patients (p=0.069). We have revised the text to clearly state that our findings did not confirm this association and emphasize the need for further research. (Manuscript: Discussion, Line 316-322)
Comment 16: Replace "is linked to a higher risk of frailty" with "is associated with frailty" to avoid implying causation.
Response: Thank you for your suggestion. We have revised the sentence to replace "is linked to a higher risk of frailty" with "is associated with frailty" to avoid implying causation.
Comment 17: The discussion mentions that reduced GFR is associated with frailty, but the study did not test GFR as an independent predictor in the logistic regression model (Table 6). The discussion could include this limitation and suggest that future analyses test GFR as an independent predictor in logistic regression to assess its specific contribution to frailty.
Suggestion: Include a comment indicating that a regression analysis to test GFR as an independent predictor of frailty could strengthen this hypothesis.
Response: Thank you for your insightful comment. We acknowledge that while our study observed an association between reduced GFR and frailty, we did not test GFR as an independent predictor in the logistic regression model (Table 6). This is a limitation of our analysis, and future studies should incorporate regression analyses to evaluate the specific contribution of GFR to frailty. (Manuscript: Discussion, Line 322-336)
Suggestion: Include a standardized regression analysis (β-coefficients) to compare the effect of frailty, CCI, and GFR on quality of life.
Response: Thank you for the suggestion. We have already incorporated this analysis in our study using hierarchical linear regression, and the standardized regression coefficients (β-coefficients) for frailty, CCI, and GFR are presented in Tables 5 and 6. (Manuscript: Result, Line 244,245)
Comment 18: The study did not test whether hypoalbuminemia causes frailty; it only found lower albumin levels in frail patients. The phrase suggests that hypoalbuminemia leads to frailty progression, but there are no longitudinal data to confirm this causal relationship.
Response: Thank you for the observation. We have revised the discussion to clarify that our findings indicate a correlation rather than causation. (Manuscript: Discussion, Line 298-299, 308-313)
Conclusion
Comment 19: This sentence suggests that having better PCS and MCS protects against frailty, which was not tested in the study (since it does not have a longitudinal design). The study only found a negative association between frailty and HRQOL but cannot claim that better HRQOL reduces the risk of frailty.
Response: Thank you for your observation. We acknowledge that our study design is cross-sectional, which limits our ability to establish a causal relationship between HRQOL and frailty. We have revised the sentence to clarify that our findings indicate an association rather than a protective effect. (Manuscript: Conclusion, Line 381)
Comment 20: The use of "the independent predictor" suggests that frailty was the only predictor, which was not demonstrated in the study.
Suggestion: "Frailty was an independent predictor of reduced physical and mental health in CKD patients."
Response: We appreciate the reviewer's observation. The phrase "the independent predictor" has been modified to avoid the implication that frailty was the sole predictor. (Manuscript: Conclusion, Line 382-383)
Comment 21: The study did not analyze frailty progression, as it is cross-sectional. It cannot claim that treatments modify frailty progression, as this requires longitudinal studies or clinical trials. This statement could be reformulated to emphasize the importance of future research on effective interventions.
Suggestion: "Our study highlights the importance of addressing existing knowledge gaps on frailty in CKD and exploring potential therapeutic strategies. However, longitudinal studies are necessary to assess frailty progression and identify effective interventions to mitigate it."
Response: Thank you for your insightful feedback. We acknowledge that our study's cross-sectional design does not allow for conclusions regarding frailty progression or the impact of treatment interventions over time. To address this, we have revised the statement to avoid implying a causal relationship and instead emphasize the need for future longitudinal studies. (Manuscript: Conclusion, Line 388-391)

Round 2
Reviewer 1 Report
Comments and Suggestions for Authors
All requested tasks have bene solved
Author Response
Comment: All requested tasks have bene solved.
Response: Thank you for your feedback. We appreciate the review and are glad that all requested tasks have been satisfactorily addressed.

Reviewer 2 Report
Comments and Suggestions for Authors
Dear Authors,
While the scientific content has improved, the manuscript would still benefit from a thorough English language revision by a native speaker or professional editor, to improve clarity, fluency, and grammatical accuracy throughout.
Abstract.
Comment: The phrase “Identifying frailty early may allow for interventions to improve HRQOL...” suggests that interventions improve HRQOL, which was not tested in this study.
Comment: The expression “may allow for interventions to improve” implies an effective therapeutic pathway, which cannot be inferred from a cross-sectional study design.
Comment. The title of Table 6 and parts of the text still use the term "trajectory" ambiguously.
Suggestion: I suggest replacing it with “KDQOL-36 domains,” which is the most widely used term in the international literature. The word "trajectory" implies longitudinal evolution, which is not consistent with the cross-sectional design of the study.
Lines 357–360
Comment. The authors state that the KDQOL-36 “does not identify CKD as a determinant of frailty,” but this wording is conceptually inadequate. The KDQOL-36 was developed to assess health-related quality of life (HRQOL) in patients with CKD and is not intended to measure frailty or identify its determinants. This sentence mistakenly suggests that the instrument should detect or explain frailty based on CKD, which exceeds its original purpose.
Suggestion: Rephrase as: “Frailty is not directly assessed by the KDQOL-36, which limits the integrated analysis between physical functioning and quality of life.”
Lines 365–366
Comment. The sentence “no prior study from India has established the influence of frailty on HRQOL in CKD patients...” uses the term “established the influence,” which may be interpreted as implying causality. Given that the study has a cross-sectional design, it is not possible to establish influence or a direct effect.
Suggestion: Replace with “examined the association” or “investigated the relationship,” which more accurately reflects the observational nature of the study.
Lines 382–384
Comment. The sentence “early detection and management of frailty could potentially improve HRQOL” contains language that implies causality and therapeutic effect, which is not compatible with the cross-sectional design of the study. The verb “improve” suggests a benefit from interventions that were not evaluated.
Suggestion: Rephrase to emphasize the observed association and the need for longitudinal studies. Example: “The association between frailty and poorer HRQOL highlights the importance of considering frailty in clinical practice, although longitudinal studies are necessary to evaluate the effects of interventions.”
Lines 385–389
Comment. The statement that “nutritional therapy may play a critical role in reducing frailty and improving quality of life” implies a therapeutic effect that was not tested in this study. Although the association between malnutrition and frailty was observed, the study did not assess nutritional interventions.
Suggestion: Rephrase as a recommendation for future investigations. Example: “Future studies should investigate the potential role of nutritional therapy in reducing frailty and improving quality of life in patients with CKD.”
Conclusion
Lines 391–392
Comment. The sentence “patients with higher scores of symptom burden, PCS, and MCS had a significantly lower likelihood of being frail” implies an inverted causality. The study data show that frail patients had lower HRQOL scores, not that higher scores protect against frailty.
Suggestion: Rephrase to accurately reflect the observed association. Example: “Patients classified as frail presented significantly lower scores in the symptom burden, PCS, and MCS domains of HRQOL.”
Lines 396–398
Comment. The sentence “targeted interventions addressing modifiable risk factors may help to mitigate its adverse effects” suggests a potential benefit from interventions that were not evaluated in the study. Although it refers to modifiable factors such as nutrition and physical activity, this conclusion implies therapeutic effects not tested in a cross-sectional design.
Suggestion: Rephrase to indicate that such interventions should be investigated in future studies. Example: “Future studies should assess whether interventions targeting modifiable risk factors, such as nutrition and physical activity, can mitigate the adverse effects of frailty on quality of life.”
Line 399
Comment. The sentence “our study underscores the importance of addressing the existing knowledge gap” is generic and not very informative. It does not clearly specify which gap was addressed by the study, which may weaken the scientific rationale.
Kind regards.
Comments on the Quality of English Language
I recommend a thorough language revision by a native English speaker or a professional scientific editor, as the manuscript contains several grammatical and stylistic inconsistencies that may affect clarity and readability.
Author Response
Abstract.
Comment: The phrase “Identifying frailty early may allow for interventions to improve HRQOL...” suggests that interventions improve HRQOL, which was not tested in this study.
Response: Thank you for your insightful comment. To clarify, we have revised the sentence to avoid implying a causal relationship.
Comment: The expression “may allow for interventions to improve” implies an effective therapeutic pathway, which cannot be inferred from a cross-sectional study design.
Response: Thank you for your valuable feedback. To address this concern, we have revised the statement to avoid implying a causal relationship.
Comment. The title of Table 6 and parts of the text still use the term "trajectory" ambiguously.
Suggestion: I suggest replacing it with “KDQOL-36 domains,” which is the most widely used term in the international literature. The word "trajectory" implies longitudinal evolution, which is not consistent with the cross-sectional design of the study.
Response: Thank you for your suggestion. To improve clarity and align with standard terminology, we have replaced “trajectory” with “KDQOL-36 domains” in the title of Table 6 and the relevant sections of the text.
Lines 357–360
Comment: The authors state that the KDQOL-36 “does not identify CKD as a determinant of frailty,” but this wording is conceptually inadequate. The KDQOL-36 was developed to assess health-related quality of life (HRQOL) in patients with CKD and is not intended to measure frailty or identify its determinants. This sentence mistakenly suggests that the instrument should detect or explain frailty based on CKD, which exceeds its original purpose.
Suggestion: Rephrase as: “Frailty is not directly assessed by the KDQOL-36, which limits the integrated analysis between physical functioning and quality of life.”
Response: Thank you for your insightful comment. We acknowledge the conceptual inaccuracy in our original wording. To ensure clarity and align with the intended purpose of the KDQOL-36, we have revised the sentence.
Lines 365–366
Comment: The sentence “no prior study from India has established the influence of frailty on HRQOL in CKD patients...” uses the term “established the influence,” which may be interpreted as implying causality. Given that the study has a cross-sectional design, it is not possible to establish influence or a direct effect.
Suggestion: Replace with “examined the association” or “investigated the relationship,” which more accurately reflects the observational nature of the study.
Response: Thank you for your valuable feedback. We acknowledge that our study's cross-sectional design does not allow for causal inferences. To ensure accuracy, we have revised the sentence.
Lines 382–384
Comment: The sentence “early detection and management of frailty could potentially improve HRQOL” contains language that implies causality and therapeutic effect, which is not compatible with the cross-sectional design of the study. The verb “improve” suggests a benefit from interventions that were not evaluated.
Suggestion: Rephrase to emphasize the observed association and the need for longitudinal studies. Example: “The association between frailty and poorer HRQOL highlights the importance of considering frailty in clinical practice, although longitudinal studies are necessary to evaluate the effects of interventions.”
Response: Thank you for your insightful feedback. To ensure clarity and accuracy, we have revised the sentence in the manuscript.
Lines 385–389
Comment: The statement that “nutritional therapy may play a critical role in reducing frailty and improving quality of life” implies a therapeutic effect that was not tested in this study. Although the association between malnutrition and frailty was observed, the study did not assess nutritional interventions.
Suggestion: Rephrase as a recommendation for future investigations. Example: “Future studies should investigate the potential role of nutritional therapy in reducing frailty and improving quality of life in patients with CKD.”
Response: Thank you for your valuable feedback. As per suggestion we have revised the sentence.
Conclusion
Lines 391–392
Comment: The sentence “patients with higher scores of symptom burden, PCS, and MCS had a significantly lower likelihood of being frail” implies an inverted causality. The study data show that frail patients had lower HRQOL scores, not that higher scores protect against frailty.
Suggestion: Rephrase to accurately reflect the observed association. Example: “Patients classified as frail presented significantly lower scores in the symptom burden, PCS, and MCS domains of HRQOL.”
Response: Thank you for your insightful comment. We acknowledge that the original wording may imply an inverted causality. To accurately reflect the observed association, we have revised the sentence.
Lines 396–398
Comment: The sentence “targeted interventions addressing modifiable risk factors may help to mitigate its adverse effects” suggests a potential benefit from interventions that were not evaluated in the study. Although it refers to modifiable factors such as nutrition and physical activity, this conclusion implies therapeutic effects not tested in a cross-sectional design.
Suggestion: Rephrase to indicate that such interventions should be investigated in future studies. Example: “Future studies should assess whether interventions targeting modifiable risk factors, such as nutrition and physical activity, can mitigate the adverse effects of frailty on quality of life.”
Response: We appreciate the reviewer’s feedback. To address this concern, we have revised the statement to emphasize the need for future research.
Line 399
Comment: The sentence “our study underscores the importance of addressing the existing knowledge gap” is generic and not very informative. It does not clearly specify which gap was addressed by the study, which may weaken the scientific rationale.
Response: Thank you for your valuable feedback. We recognize that there were repetitions regarding the need for future studies, which were not necessary multiple times. To maintain clarity and consistency, we have removed the redundant statement while ensuring that the discussion remains focused on the study’s key contributions.
